# Locating Charging Stations of Various Sizes with Different Numbers of Chargers for Battery Electric Vehicles

**Shaohua Cui [1] , Hui Zhao [1,2,]*  and Cuiping Zhang [3] **

[1]  MOE Key Laboratory for Urban Transportation Complex System Theory and Technology, School of Traffic and Transportation, Beijing Jiaotong University, Beijing 100044, China; 16120747@bjtu.edu.cn

[2]  Key Laboratory of Transport Industry of Big Data Application Technologies for Comprehensive Transport, Beijing Jiaotong University, Beijing 100044, China

[3]  School of Computer Science, Beijing Information Science & Technology University, Beijing 100192, China; cpzhang@bistu.edu.cn

*  Correspondence: zhaoh@bjtu.edu.cn

**Abstract:** Compared with traditional fuel vehicles, battery electric vehicles (BEVs) as a sustainable transportation form can reduce carbon dioxide emissions and save energy, so its market share has great potential. However, there are some problems, such as: Their limited range, long recharging time, and scarce charging facilities, hindering improvement in the market potential of BEVs. Therefore, perfect and efficient charging facility deployment for BEVs is very important. For this reason, the optimal locations for charging stations for BEVs are investigated in this paper. Instead of flow-based formulation, this paper is based on agents under strictly imposed link capacity constraints, where all agents can select their routes and decide on the battery recharging plan without running out of charge. In our study, not only the locations of charging stations, but also the size of charging stations with the different number of chargers, would be taken into consideration. Then, this problem is formulated as a location problem for BEV charging stations of multiple sizes based on agents under link capacity constraints. This problem is referred to as the agent-refueling, multiple-size location problem with capacitated network (ARMSLP-CN). We formulate the ARMSLP-CN as a 0–1 mixed-integer linear program (MILP) with the aim to minimize the total trip time for all agents, including four parts, namely, the travel time, queue time, fixed time for recharging, and variable recharging time depending on the type of charger and the amount of power recharged, in which commercial solvers can solve the linearized model directly. To demonstrate this model, two different numerical instances are designed, and sensitivity analyses are also presented.

**Keywords:** battery electric vehicles; charging station location; charging station size; agent-based model; recharging

## 1. Introduction

Based on environmental and economic concerns, the deployment of BEVs has increased significantly in recent years [1–4]. Compared with traditional gasoline vehicles (GVs), BEVs have many advantages, such as low or no environment emissions, and energy efficiency, to make them an ideal form of transport for urban areas [5]. Lee and Han [6] estimated that there will be an annual growth rate of more than 25% for the sales of BEVs by 2025. The popularity of BEVs is not only due to its environmental friendliness, but also due to various incentives provided by governments, such as purchase incentives, free parking, as well as less access restrictions [7].

Though the benefits of deploying BEVs are undeniable, there is still a barrier for consumers to choose BEVs: Range anxiety [8]. Range anxiety refers to the fact that BEV drivers fear that their batteries would run out of power en route due to the limited battery power capacity [9,10]. This range anxiety makes BEV drivers reluctant to take long journeys. Furthermore, recharging a battery completely takes too much time using current recharging technology. Therefore, some studies proposed many alternative methods to avoid a long recharging time, such as battery-swapping [11] and charging lanes [12]. Due to their limited range, the increasing number of BEVs naturally raises the problem of electric charging stations. The insufficient refueling infrastructures pose a threat to the market share of BEVs [13–15]. Therefore, to address this situation, the problem of optimal locations for recharging stations is urgent.

Currently, two mainstream approaches of determining optimal locations for recharging stations are applied: Spatial approaches and flow-based approaches. For the former, the objective function is usually assumed to minimize the distance between the location of a recharging station and possible BEV drivers served by this facility [16]. For example, the p-median approach is one of spatial approaches where a BEV driver will recharge its battery before the power is exhausted by visiting recharging stations, while the aim is to minimize the number of stations and maximize coverage at the same time. Therefore, this method of deploying the location of stations is named as the spatial approach. However, it is impossible for complete coverage across a region, so this method can only be conducted to solve urban transport planning problems [17]. The second approach inserts recharging stations into traffic routes to make BEV travel each segment with sufficient power and only recharge its battery in each junction between two adjacent segments, which is named as a flow refueling location model (FRLM) by Kuby and Lim [18]. In general, the approach of FRLM is widely used to deploy charging stations [18,19].

These models based on the FRLM pattern presented above need to enumerate all recharging patterns. However, when the number of OD pairs is huge, considering all possible paths is already an enormous challenge; generating all feasible patterns for each path will result in huge batteries even before the calculation of a model. A mathematical formulation was proposed by He et al. [8] in which the recharging decision variables are embedded. In the same way, in our research, considering the time-consuming nature of path enumeration, we record the charge states of a BEV battery through the variables and equations presented by He et al. [8], avoiding the full enumeration of paths and recharging patterns.

Furthermore, all previous research on the location problem of charging stations has been based on flow. However, modeling frameworks based on non-atomic and atomic games are different. The traditional flow model with non-atomic players does not show crowding between agents, so the agent-based model is utilized in this paper. In addition, the relevant models for electric vehicles, which need to track their charge levels, are quite different from those of traditional fuel vehicles. Different initial charge levels and anxiety ranges will result in different recharging patterns or even entirely different routes for drivers. However, traditional flow models, such as user equilibrium (UE), stochastic user equilibrium (SUE), and so on, assume that all BEV drivers have the same initial charge level and anxiety range, sometimes assuming different OD pairs have different initial charge levels or anxiety ranges, but they do not reflect the differences between the agents of one OD pair. In reality, there are many people who arrive at the same destination from the same point of departure but with differences in the initial charge levels and anxiety ranges. However, traditional flow models do not reflect these differences, so the agent-based model is a better choice. In this paper, an agent-based refueling location model is proposed to design the location of recharging stations.

Furthermore, recharging time is considered sufficiently in this paper, because when the amount of power reaches 80% of the battery capacity, the recharging rate is significantly reduced [20]. In some studies, such as Nie and Ghamami [21], they assumed that the fast-charging devices are deployed to ignore recharging time, but these devices are very expensive and require high voltages, which make large-scale deployment of such charging devices difficult. To avoid high construction costs, multiple types of charging stations with different charging rates were considered by Wang and Lin [22].

However, in the study, the number of chargers is not studied. In reality, charging stations of different sizes with different numbers of chargers may greatly affect the service efficiency. Therefore, this paper optimizes the combination of charging stations of different sizes under a given total budget. Specifically, recharging time includes three parts, named queue time, fixed charging activity preparation time, and variable charging time related to the amount power recharged and type of charger. The queue time, in this paper, is only a generalized time and just conducted to reflect the difference of charging stations with different numbers of chargers. Specifically, the queue time will reduce with the increase of chargers.

Specifically, this paper investigates the location problem for multiple sizes of BEV charging stations based on agents under a link capacity constraint. Hereafter, our approach is defined as the agent-refueling, multiple-size location problem with capacitated network (ARMSLP-CN). Furthermore, we formulate the ARMSLP-CN as a 0–1 MILP with the aim of minimizing the total trip time. To the best of our knowledge, until now, there has been no research focusing on the multiple-size recharging station location problem. The contributions of this paper can be described as follows.

(1)    The refueling station location problem based on agents is proposed for the first time.
(2)    We are the first to discuss the multiple-size refueling station location problem with various numbers of chargers.
(3)    The recharging time based on the amount of energy recharged and the size of the charging stations are considered.
(4)    We present a linear mathematical formulation for the ARMSLP-CN considering both limited operational ranges and possible recharging strategies.

For the remainder, the problem is formally introduced in Section 2. The MILP formulation of the ARMSLP-CN is shown in the next section. The computational results and sensitivity analyses are presented and discussed in Section 4. We conclude the study in Section 5.

## 2. Problem Description

For a more detailed and clear discussion of the multiple-size refueling station location problem based on agents, this problem is further described in this section. Specifically, Section 2.1 lists all the notations used in this paper for convenience. Then, some reasonable assumptions are proposed in Section 2.2. Section 2.3 describes the three parts of the charging time for a BEV at a charging station in detail. Finally, in Section 2.4, the major features of ARMSLP-CN model are presented through a simple numerical example.

*2.1. Notations*

The general indices, sets, parameters, and variables appearing in the remainder are listed in Table 1.

**Table 1.** Indices, sets, parameters, and variables.

| Indices | Definition |
|---|---|
| $i, j$ | Index of nodes, $i, j \in N$ |
| $(i, j)$ | Index of physical link between two adjacent nodes, $(i, j) \in L$ |
| $a$ | Index of agents, $a \in A$, defined based on each OD pair |
| $w$ | Index of OD pair, $w \in W$ |
| $o(a)$ | Index of origin node of agent $a$ |
| $d(a)$ | Index of destination node of agent $a$ |
| **Sets** | |
| $N$ | Set of nodes in the physical transportation network |
| $L$ | Set of links |
| $W$ | Set of origin–destination (OD) pairs |
| **Parameters** | |

**Table 1.** *Cont.*

| Indices | Definition |
|---|---|
| $d_{ij}$ | Distance of link $(i, j) \in L$ |
| $t_{ij}$ | Travel time of link $(i, j) \in L$ |
| $Cap_{ij}$ | Capacity of link $(i, j) \in L$ |
| $c^1$ | Fixed time for recharging |
| $c^2$ | Recharging rate, which depends on the type of charger |
| $c^3$ | Transfers the gap between the charger number of one node and maximum charger capacity as the queuing time |
| $cf^f$ | Fixed cost of a station |
| $cf^v$ | Cost of a charger |
| $B$ | Total financial budget |
| $M, K, Q$ | Assumed large value as auxiliary parameters |
| $m^a$ | Range anxiety value for agent $a$ |
| $\overline{w}$ | Energy consumption rate of BEVs |
| $z_{max}$ | Maximum charger capacity |
| $z_{min}$ | Minimum number of chargers |
| $L_{max}$ | Battery size |
| $L_0$ | Initial charge level |

| Variables | |
|---|---|
| $x^a_{i,j}$ | $= 1$ if agent $a$ is assigned to link $(i, j) \in L$; $= 0$ otherwise |
| $s\_t^a_i$ | Total charging service time for agent $a$ at node $i \in N$, i.e., $s\_t^a_i = c^1 + c^2 F^a_i$ |
| $q\_t^a_i$ | Queuing time depends on the gap between the number of chargers $z_i$ at node $i \in N$ and the maximum charger capacity $z_{max}$ of one node |
| $r^a_i$ | $= 1$ if agent $a$ recharges at node $i \in N$; $= 0$, otherwise |
| $F^a_i$ | Amount of electricity recharged at node $i \in N$ for agent $a$ |
| $g^a_i$ | Transition variable, equal to the gap between $z_i$ and $z_{max}$ when agent $a$ recharges at node $i$, otherwise, $g^a_i$ equals zero. |
| $s_i$ | $= 1$ if the recharging station is built at node $i \in N$; $= 0$, otherwise |
| $z_i$ | Number of chargers at node $i \in N$ |
| $L^a_i$ | Charge level at node $i$ after recharging agent $a$ |
| $\rho^a_{ij}$ | $= 0$ if link $(i, j)$ is utilized; unrestricted otherwise for agent $a$ |

*2.2. Basic Assumptions*

In recent years, several studies have been conducted on charging station location problems for BEVs. Wirges et al. [23] developed a spatial-temporal model (STM) based on an economic model to finance charging infrastructures of BEVs for the city of Stuttgart, Germany. Another STM was developed by Tu et al. [24] to optimize charging station locations of an electric taxi fleet using the data of Shenzhen, China. The location problem of multiple types of charging devices providing different charging rates was studied by Wang and Lin [22]. Capar et al. [25] developed the FRLM to make it cover all arcs of each path. Chen et al. [12] introduced a model to deploy charging lanes. Strehler et al. [26] mixed BEVs and hybrid vehicles, and constructed a shortest path problem based on the FRLM. Hof et al. [27] focused on the battery swap station location and established this problem on electric vehicle routing problems (E-VRPs). There were also studies in which SUE was used to optimize charging stations, such as those by Hanabusa and Horiguchi [28], Riemann et al. [29], and so on. Xylia et al. [30] used the Geographic Information System (GIS) software ArcGIS to optimize the location of the charging devices for electric buses. These developed macroscopic models cannot optimize the specific sizes of charging stations. However, the number of chargers greatly affects the service efficiency of charging stations, so there is a huge meaning to characterize the sizes of charging stations. Therefore, before formulating the model, the following assumptions are made.

(I) The link is flat, i.e., no grades are considered, so the travel time between each node is constant and given.

(II)　Each link has its own capacity. In other words, the sum of agents passing through one link is less than or equal to the capacity of this link and will not exceed it.

(III)　A linear recharge is assumed, which means the recharging rate is a fixed constant depending only on the level of the charger, and the type of charger used in each instance is known in advance.

(IV)　Charging stations of different sizes use the same type of charger; only the number of chargers is different. Therefore, the recharging rates of the different sizes are the same, and only the queuing time differs. The queuing time decreases as the size of the charging station increases.

Gradient and overcrowded links significantly affect a BEV's speed. Some conditions, such as weather, temperature, speed, may change the travel ranges of BEVs up to 60% [31]. Straubel [32] reported that the travel range of Tesla, a maker of BEVs, may be reduced by half when the driver doubles the speed. Therefore, the uncertain travel range is a complex study where Lee and Han [6] established the problem in a probabilistic consideration, so assumptions (I) and (II) are proposed to simplify the problem. In addition, the charging rate may increase significantly for the last 10 to 20% of the battery capacity [20]; thus, for simplicity, (III) is assumed. Liu and Wang [33] studied the location problem of multiple types of recharging facilities which can provide different recharging speeds, and the recharging speed also depends on the recharging equipment. Finally, the different sizes are the main research focus and the type of charger is determined in advance, so assumption (IV) is reasonable.

*2.3. Calculation of Charging Time*

To describe the charging time more accurately, in this section, the three parts of the charging time will be introduced separately, and the calculation method will also be described. In Section 1, the charging time is divided into three parts, which consist of the queuing time, the fixed recharging preparation time, and the variable time. The latter two can be collectively referred to as the service time.

2.3.1. Calculation of Queuing Time

For the accurate calculation of the queue time, it is necessary to determine the number of agents at the charging station and the amounts of energy to be charged by the agents who are charging at that moment. While fully recharging electric cars takes a long time, sometimes even more than two hours, in reality electric car drivers will not wait for more than two hours. Therefore, in this paper, we ensure that the queue time for larger charging stations is shorter, which is a generalized time. Let $q\_t_i^a$ be the queuing time depending on the gap between the number of chargers $z_i$ at node $i \in N$ and the maximum charger capacity $z_{max}$ of one node, which means that when the number of chargers of one station is $z_{max}$, the queuing time is not considered. In this paper, the queuing time is only a generalized time, just to reflect the queuing time difference of charging stations with different numbers of chargers. If the queuing time of a charging station with the maximum charger $z_{max}$ is considered, a constant can be added. Here, $c^3$ is a parameter to transfer the gap as the queuing time. For simplicity, a transition variable $g_i^a$ is defined. We assume that if agent $a$ recharges at node $i$ with the charger, then $g_i^a$ equals the gap between $z_i$ and $z_{max}$; otherwise, $g_i^a$ equals zero. Furthermore, the equation of the queuing time can be written as $q\_t_i^a = c^3 g_i^a$. If one charging station $i$ is chosen for recharging by agent $a$, then the queuing time function is $q\_t_i^a = c^3 g_i^a = c^3(z_{max} - z_i)$; otherwise, $q\_t_i^a = 0$. More formulae are dealt with in Section 3.

2.3.2. Calculation of Service Time

Let $s\_t_i^a$ denote the total charging service time taken by BEV agent $a$ to recharge an $F_i^a$ amount of electricity at node $i \in N$. In Section 2.2, a linear recharge has been assumed, so the charging function can be described as $s\_t_i^a = c^1 + c^2 F_i^a$. In this function, the first part $c^1$ is fixed recharging preparation time, and the second part calls variable time, where $c^2$ represents the recharge rate depending only on the type of charger, shown in Table 2 in detail. The type of charger is determined in advance, and all

stations use the same type of charger. Clearly, both $c^1$ and $c^2$ are equal to zero when a charging station is not built at node $i \in N$.

**Table 2.** BEV charger specifications [8].

| Charging Level | Level 1 | Level 2 | Level 3 |
|---|---|---|---|
| Power (kW) | 1.44 | 6 | 90 |
| Charging circuit | 120 V, 15 A | 240 V, 30 A | 500 V, 200 A |
| Fixed time for recharging activity (min) | 5 | 5 | 5 |
| Recharge rate (min/kWh) | 41.67 | 10 | 0.67 |

### 2.4. A Simple Example

Before a simple example is presented, the definition of a usable path for BEVs is explained. In comparison with fuel vehicle drivers, BEV drivers are often anxious about the limited battery capacity, which is usually referred as range anxiety. To characterize the route choice behavior for BEV drivers, He et al. [8] defined the concept of a usable path, which is a path that a BEV driver could complete with or without en-route charging.

To explain charging time clearly, a numerical network from He et al. [8] is presented. The network is shown in Figure 1, with four nodes and five links along with one OD pair 1–4. The link capacity $Cap_{ij}$, travel time $t_{ij}$, and length $d_{ij}$ parameters for each link are illustrated in the figure. On the right of the figure, three paths for the OD pair 1–4 are listed. The demand for the OD pair 1–4 is 2, and range anxieties $m^a$ are equal to zero. The data related to the BEV are summarized as follows. The battery capacity is $L_{max}$ = 18 kWh, the initial charge is $L_1$ = 6 kWh, and the energy consumption rate $\overline{w}$ is 2 kWh/mi. Nodes 2 and 3 are charging stations with *Level* 3 rechargers where the BEV can be recharged. Therefore, from Table 2, we find that the fixed time $c^1$ is equal to 5 min, and the recharge rate $c^2$ is equal to 0.67 min/kWh. We assume that the charger number $z_2$ is 4 char at node 2 ("4 char" means that there are four chargers built), and that of $z_3$ is 3 char at node 3. However, the charger number limit $z_{max}$ is 5 char, which is reasonable because the construction costs of all charging stations must be under a certain budgeted amount. Thus, it is possible that the number of chargers in all the charging stations is lower than the maximum capacity.

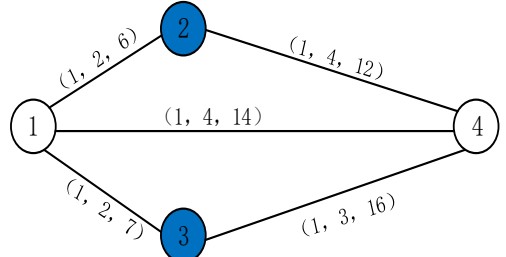

| Path ID | Node Sequence |
|---|---|
| Path 1: | 1→2→4 |
| Path 2: | 1→4 |
| Path 3: | 1→3→4 |

(link capacity (veh/h), distance (mi), travel time (min))

**Figure 1.** A numerical network presented in He et al. [8].

The initial charge of the BEV is $L_1$ = 6 kWh, and the energy consumption rate $\overline{w}$ is 5 kWh/mi. Thus, the BEV can run 3 mi with no en-route charging. Therefore, paths 1 and 3 are both usable, where the BEV can be recharged at node 2 and node 3, respectively, and path 2 is not usable. From Figure 1, we can calculate that the length of path 1 is 6 mi and that of path 5 mi. The total electricity consumption to travel path 1 is 12 kWh, i.e., 6 × 2 = 12 kWh. Similarly, the total consumption is 10 kWh to travel path 3, i.e., 5 × 2 = 10 kWh. Due to the link capacity, paths 1 and 3 can be traveled by one agent, where we assume that agent 1 chooses path 1 and agent 2 chooses path 2, owing to the same characteristics of these two agents, such as range anxiety, initial charge level, and battery capacity.

Therefore, the amount of electricity to be recharged at node 2 is $F_2^1 = 12 - 6 = 6$ kWh, and similarly, $F_3^2 = 10 - 6 = 4$ kWh. Then, the queuing time for these two agents can be respectively calculated as

$$q\_t_2^1 = c^3 g_2^1 = c^3(z_{max} - z_2) = 1 \times (5 - 4) = 1 \text{ min} \tag{1}$$

$$q\_t_3^2 = c^3 g_3^2 = c^3(z_{max} - z_3) = 1 \times (5 - 3) = 2 \text{ min} \tag{2}$$

where $c^3$ equals 1 min/char. Figure 1 is just an example network with only two drivers. Each driver selects a separate route, so only one driver charges a vehicle per charging station. The number of charging chargers is greater than the number of drivers who need to charge a vehicle, so it seems irrational to add a queuing time to each driver's travel time. However, as described in Section 2.3.1, the queuing time in this paper is only a generalized time because, in a real network, the number of chargers for a charging station is smaller than the number of drivers with charging requirements. This example shown in Figure 1 is only used to introduce the method for calculating queuing time.

The service time for these two agents can be calculated as follows:

$$s\_t_2^1 = c^1 + c^2 F_2^1 = 5 + 0.67 \times 6 = 9.02 \text{ min} \tag{3}$$

$$s\_t_3^2 = c^1 + c^2 F_3^2 = 5 + 0.67 \times 4 = 7.68 \text{ min} \tag{4}$$

Then, the total trip time of agent 1 can be calculated as

$$t_{12} + q\_t_2^1 + s\_t_2^1 + t_{24} = 6 + 1 + 9.02 + 12 = 28.02 \text{ min} \tag{5}$$

Similarly, the total trip time of agent 2 is calculated as

$$t_{13} + q\_t_3^2 + s\_t_3^2 + t_{34} = 7 + 2 + 7.68 + 16 = 32.68 \text{ min} \tag{6}$$

Through the above example, our problem is further explained and then we will propose the mathematical model in the next section.

## 3. Model Formulation

We build the model on a directed graph with $N$ as the set of nodes and $L$ as the set of links. Each agent $a$ has its own origin $o(a)$, destination $d(a)$, and comfortable electricity range $m^a$. Each link $(i, j)$ has a travel time $t_{ij}$, distance $d_{ij}$, and capacity $Cap_{ij}$. To improve the efficiency of the total transportation system, a total budget $B$ is assumed. The fixed cost of a new charging station is $cf^f$, and the cost of a single charger is $cf^v$. Let the binary variable $s_i$ indicate whether a charging station is constructed at node $i$, and let non-negative integer variable $z_i$ represent the number of chargers built, where the lower and upper bounds are $z_{min}$ and $z_{max}$, respectively. For an agent $a$ traveling between the OD pair $w \in W$, the amount of electricity recharged at node $i$ is $F_i^a$, and the charge level at node $i$ after recharging is $L_i^a$, where $L_{max}$ and $L_0$ represent the battery capacity and the initial amount of power. Let $r_i^a$ indicate whether an agent $a$ recharges at node $i$. Some sufficiently large constants are defined, such as $M, K, Q$. Let $\overline{w}$ be the energy consumption rate. Here, the binary variable $x_{ij}^a$ equals 1 when link $(i, j)$ is traveled and 0 otherwise. Further, variable $\rho_{ij}^a$ equals 0 when link $(i, j)$ is traveled and is unlimited otherwise.

ARMSLP-CN:

$$\min \sum_a \sum_{(i,j)} t_{ij} x_{ij}^a + \sum_a \sum_i (c^1 r_i^a + c^2 F_i^a + c^3 g_i^a) \tag{7}$$

Subject to

Flow balance:

$$\sum_{i:(i,j)\in L} x_{ij}^a - \sum_{i:(j,i)\in L} x_{ji}^a = \begin{cases} -1, & j = o(a) \\ 1, & j = d(a) \\ 0, & \text{otherwise} \end{cases} , \forall a \tag{8}$$

Budget constraint:

$$\sum_{i\in N} \left(cf^f s_i + cf^v z_i\right) \leq B, \forall i \in N \tag{9}$$

Station capacity constraint:

$$z_i \leq z_{max} s_i, \forall i \in N \tag{10}$$

$$z_i \geq z_{min} s_i, \forall i \in N \tag{11}$$

Link capacity constraint:

$$\sum_a x_{ij}^a \leq Cap_{ij}, \forall (i,j) \in L \tag{12}$$

Electricity constraint:

$$L_j^a - L_i^a + d_{ij}\overline{w} - F_j^a = \rho_{ij}^a, \forall (i,j) \in L, \forall a \tag{13}$$

$$L_i^a - d_{ij}\overline{w} \geq -K(1 - x_{ij}^a) + m^a, \forall (i,j) \in L, \forall a \tag{14}$$

$$-M(1 - x_{ij}^a) \leq \rho_{ij}^a \leq M(1 - x_{ij}^a), \forall (i,j) \in L, \forall a \tag{15}$$

$$0 \leq F_i^a \leq Qs_i, \forall i \in N, \forall a \tag{16}$$

$$0 \leq L_i^a \leq L_{max}, \forall i \in N, \forall a \tag{17}$$

$$L_o^a = L_0 \tag{18}$$

$$r_i^a \geq \frac{F_i^a}{Q}, \forall i \in N, \forall a \tag{19}$$

Charging delay constraint:

$$g_i^a + z_i \geq r_i^a z_{max}, \forall i \in N, \forall a \tag{20}$$

Variable constraints:

$$x_{ij}^a \in \{0,1\}, \forall (i,j) \in L, \forall a \tag{21}$$

$$s_i \in \{0,1\}, \forall i \in N \tag{22}$$

$$r_i^a \in \{0,1\}, \forall i \in N, \forall a \tag{23}$$

$$z_i \in \mathbb{N}, \forall i \in N, \forall a \tag{24}$$

$$g_i^a \geq 0, \forall i \in N, \forall a \tag{25}$$

$$\rho_{ij}^a \in \mathbb{R}, \forall (i,j) \in L, \forall a \tag{26}$$

The objective Equation (7) is to minimize the total trip time of all agents, which includes three parts. The first part is the travel time $\sum_a \sum_{(i,j)} t_{ij} x_{ij}^a$; the second part is the charging time $\sum_a \sum_i \left(c^1 r_i^a + c^2 F_i^a\right)$, in which the first component is the fixed recharging preparation time and the second component represents the variable time depending on the amount of recharged electricity; and the third part presents queue time $\sum_a \sum_i c^3 g_i^a$. Equation (8) is a traditional agent-based flow balance constraint. Equation (9) is the budget constraint, where the first part is the fixed cost of a recharging station, and the second part is the variable cost, which is calculated by multiplying the number of chargers by the cost of a charger. The charger number capacity is the same as (or lower) than the maximum capacity of a station, constraint (10), and higher than the minimum constraint (11). Equation (12) represents the link capacity constraint. Equations (13) and (15) specify the relationship between the charge levels at starting and ending nodes where a link is utilized. In Equation (15),

$$M = \max\left\{ L_{max} + \max_{(i,j)\in L}(d_{ij})\cdot\overline{w}, 2L_{max} - \min_{(i,j)\in L}(d_{ij})\cdot\overline{w} \right\} + 1.$$ Equation (14) specifies that the charge level is higher than the comfortable range where $K = \max_{(i,j)\in L}(d_{ij})\cdot\overline{w} + m^a + 1$. Equation (16) suggests that a BEV only recharges its battery at nodes with a charging station where $Q = L_{max} + 1$. Equation (17) sets the upper and lower bounds of the charge levels. Equation (18) sets the initial charge level. Equation (19) determines if agent $a$ chooses to recharge his or her battery at node $i$. Equation (20) calculates the queue time. Equations (21)–(23) specify the binary variables $x_{ij}^a$, $s_i$, and $r_i^a$, respectively. Equation (24) specifies that the variable $z_i$ is a non-negative integer. Equation (25) specifies that the variable $g_i^a$ is a non-negative real number. Equation (26) specifies that the variable $\rho_{ij}^a$ is a real number.

For the reader's convenience, in this part, the queue time will be further explained. In Section 2.3.1, the calculation of the queue time is described in words, where if one charging station $i$ is chosen for recharging by agent $a$, then the queue time function is $q\_t_i^a = c^3 g_i^a = c^3(z_{max} - z_i)$; otherwise, $q\_t_i^a = 0$. This is dealt with by Equation (20) where $r_i^a$ indicates whether the agent $a$ recharges at node $i$. If $r_i^a$ equals 1, which means agent $a$ recharges at node $i$, then Equation (20) can be rewritten as

$$g_i^a + z_i \geq z_{max} \tag{27}$$

Equation (27) can be further reformulated as

$$g_i^a \geq z_{max} - z_i \tag{28}$$

Since the objective is to minimize $\sum_a \sum_i c^3 g_i^a$, $g_i^a$ takes the value $(z_{max} - z_i)$. If $r_i^a$ equals 0, where agent $a$ would not recharge at node $i$, Equation (20) can be reformulated as

$$g_i^a \geq -z_i \tag{29}$$

Here, $g_i^a$ is a positive real number according to Equation (25), and the minimization of $\sum_a \sum_i c^3 g_i^a$ is required, so $g_i^a$ is equal to zero. Therefore, this model can reasonably reflect the queue time.

## 4. Numerical Experiments

This section examines the presented model by two different instances. The computing device was a personal computer with an Intel(R) Core(TM) i7 6700U 3.40 GHz CPU and 16.00 GB of RAM using the Microsoft Windows 7 (64 bit) OS. To solve the linearized model, the optimization package GAMS with CPLEX 12.8 solver was employed in all networks.

The section is organized as follows. After a simple description of the Nguyen-Dupius network in Section 4.1, sensitive analyses, such as for range anxiety, initial charge level, and total cost budget, are also performed on the Nguyen-Dupius network described in Sections 4.2–4.4, respectively. In Sections 4.2 and 4.3, instead of homogeneous agents, the advantages of the agent-based model are presented. In Section 4.5, a Sioux Falls network with eight OD pairs is presented to show the computational efficiency of the model and its potential for application to real networks.

### 4.1. A Simple Case

In this section, the Nguyen–Dupius network (see Figure 2), including 13 nodes, 19 links, and four OD pairs is examined. The link capacity, distance, and travel time of each link are listed in Table 3, where the link distance is 1.5 times the link travel time [8], and the OD demand is listed in Table 4. The battery capacity $L_{max}$ is set to 24 kWh, and the energy consumption rate $\overline{w}$ is set to 0.29 kWh/mi [8]. The initial electricity level of the BEV, $L_0$, is set to 20 kWh [8]. We further assume that the total budget $B$ is 38, the fixed construction cost $cf^f$ of one charging station is 10 if no charger is built, and the construction cost of one charger $cf^v$ is 1. The maximum charger capacity $z_{max}$ is 5char, and the minimum number of chargers $z_{min}$ is 2char. For simplicity, the anxiety range values of all the agents, $m^a$ are assumed to be the same, i.e., $m^a = 2$ kWh. The BEV charger specifications are shown in Table 2 [8].

In this part, the Level 2 charging level is adopted. The equilibrium link flow is presented in Table 5, energy recharged for each OD pair is shown in Table 6, and the charging station construction program is shown in Table 7.

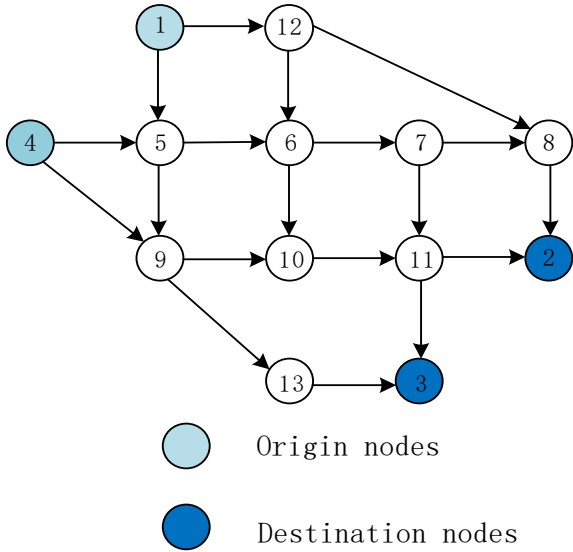

**Figure 2.** Nguyen-Dupius network.

**Table 3.** Nguyen-Dupius network information (link capacity (veh/h), distance (mi), travel time (min)).

| Link | Capacity | Distance | Travel Time | Link | Capacity | Distance | Travel Time |
|------|----------|----------|-------------|------|----------|----------|-------------|
| $(1,5)$ | 40 | 14.7 | 9.8 | $(8,2)$ | 40 | 18.9 | 12.6 |
| $(1,12)$ | 30 | 18.9 | 12.6 | $(9,10)$ | 30 | 21.0 | 14.0 |
| $(4,5)$ | 30 | 18.9 | 12.6 | $(9,13)$ | 30 | 18.9 | 12.6 |
| $(4,9)$ | 30 | 25.2 | 16.8 | $(10,11)$ | 30 | 12.6 | 8.4 |
| $(5,6)$ | 50 | 6.3 | 4.2 | $(11,2)$ | 30 | 18.9 | 12.6 |
| $(5,9)$ | 30 | 18.9 | 12.6 | $(11,3)$ | 30 | 16.8 | 11.2 |
| $(6,7)$ | 50 | 10.5 | 7.0 | $(12,6)$ | 30 | 14.7 | 9.8 |
| $(6,10)$ | 40 | 27.3 | 18.2 | $(12,8)$ | 30 | 29.4 | 19.6 |
| $(7,8)$ | 50 | 10.5 | 7.0 | $(13,3)$ | 30 | 23.1 | 15.4 |
| $(7,11)$ | 40 | 18.9 | 12.6 | | | | |

**Table 4.** OD demand.

| O/D | 2 | 3 |
|-----|-----|-----|
| 1 | 20 | 30 |
| 4 | 30 | 20 |

**Table 5.** Equilibrium link flow.

| Link | Link Flow | Link | Link Flow | Link | Link Flow |
|------|-----------|------|-----------|------|-----------|
| 0 | 30 | $(6,10)$ | | $(11,2)$ | 10 |
| $(1,12)$ | 20 | $(7,8)$ | 20 | $(11,3)$ | 30 |
| $(4,5)$ | 20 | $(7,11)$ | 30 | $(12,6)$ | 0 |
| $(4,9)$ | 30 | $(8,2)$ | 40 | $(12,8)$ | 20 |
| $(5,6)$ | 50 | $(9,10)$ | 10 | $(13,3)$ | 20 |
| $(5,9)$ | 0 | $(9,13)$ | 20 | | |
| $(6,7)$ | 50 | $(10,11)$ | 10 | | |

**Table 6.** OD Energy recharged.

| OD | *NA* | *ER* (kWh) |
|----|------|-----------|
| $1-2$ | 20 | 29.76 |
| $1-3$ | 30 | 44.64 |
| $4-2$ | 30 | 62.91 |
| $4-3$ | 20 | 29.76 |
| Total | 100 | 167.07 |

Note: "*NA*" denotes the number of agents recharged, and "*ER*" denotes the amount of energy recharged.

**Table 7.** Charging station construction program.

| Node | Charger Number (char) |
|------|----------------------|
| 5 | 4 |
| 9 | 2 |
| 12 | 2 |

Next, we will test other types of chargers, i.e., Level 1 and Level 3. The parameters of these two chargers are shown in Table 2. Figure 3 compares the total trip time of three types of chargers. It can be observed that, with increasing recharging rates, the total trip time decreases. Further, as seen in Table 8, the amount of energy recharged increases with an increasing recharging rate. As the recharging rate increases, the agents will not consider recharging longer while spending time on the link. As seen in Table 9, the network only requires two charging stations with five chargers when the *Level 3* charger is adopted, which is a good and reasonable result, indicating that when the charging technology is enhanced, less land for the construction of charging stations is required.

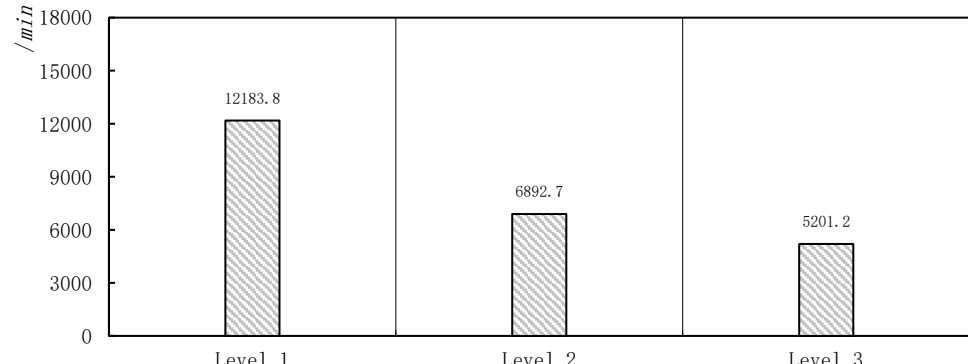

**Figure 3.** Comparison of the total trip time of the three types of charging stations.

**Table 8.** Comparison of the energy recharged for each OD pair.

| OD | Level 1 | | Level 2 | | Level 3 | |
|----|------|-----------|------|-----------|------|-----------|
| | *NA* | *ER* (kWh) | *NA* | *ER* (kWh) | *NA* | *ER* (kWh) |
| $1-2$ | 20 | 29.7 | 20 | 29.7 | 0 | 0 |
| $1-3$ | 30 | 44.6 | 30 | 44.6 | 30 | 136.0 |
| $4-2$ | 30 | 62.9 | 30 | 62.9 | 30 | 62.9 |
| $4-3$ | 20 | 29.7 | 20 | 29.7 | 20 | 29.7 |
| *Total* | 100 | 167.0 | 100 | 167.0 | 80 | 228.6 |

Notes: "*NA*" denotes the number of agents recharged, and "*ER*" denotes the amount of energy recharged.

**Table 9.** Comparison of the charging station construction programs

| Node | Level 1 | Level 2 | Level 3 |
|------|---------|---------|---------|
| 1 | 0 | 0 | 0 |
| 2 | 0 | 0 | 0 |
| 3 | 0 | 0 | 0 |
| 4 | 0 | 0 | 0 |
| 5 | 4 | 4 | 0 |
| 6 | 0 | 0 | 5 |
| 7 | 0 | 0 | 0 |
| 8 | 0 | 0 | 0 |
| 9 | 2 | 2 | 5 |
| 10 | 0 | 0 | 0 |
| 11 | 0 | 0 | 0 |
| 12 | 2 | 2 | 0 |
| 13 | 0 | 0 | 0 |

### 4.2. Various Levels of Agents' Range Anxiety

To perform the sensitivity analysis on the agents' range anxiety, five scenarios are tested to observe the impact on system performance. The system performance results of the energy recharged are listed in Figure 4, the total trip time results are shown in Figure 5, and the numbers of chargers built at each node are shown in Table 10.

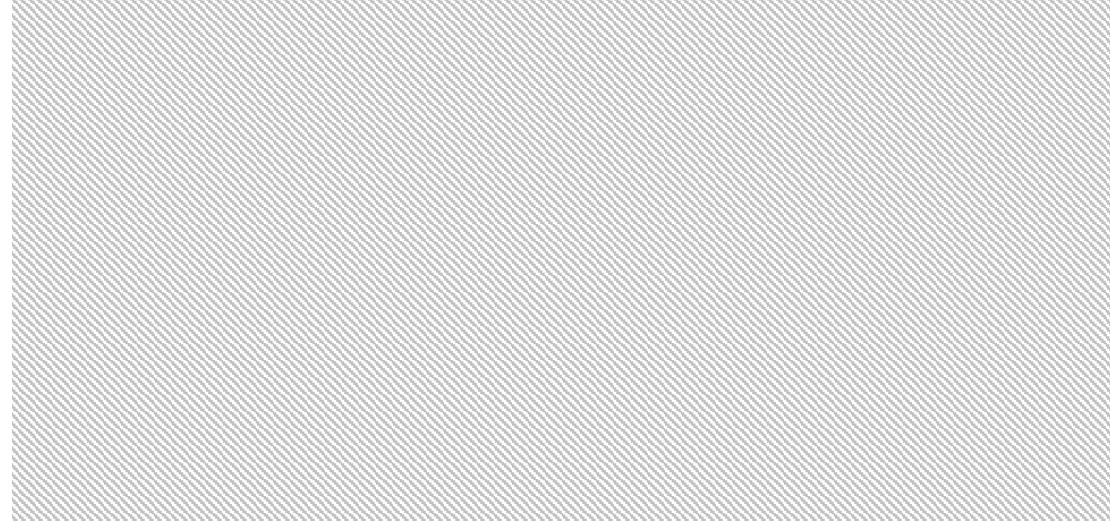

**Figure 4.** Comparison of the energy recharged for various range anxiety levels.

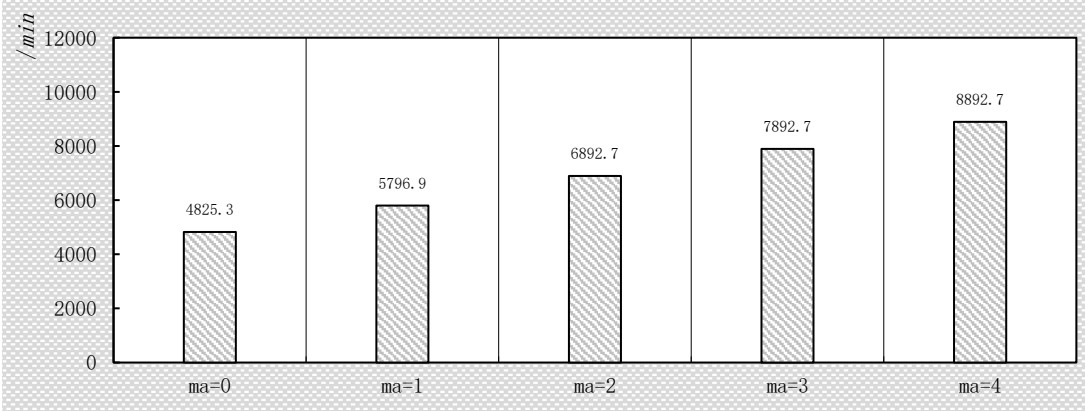

**Figure 5.** Comparison of the total trip time for various range anxiety levels.

**Table 10.** Number of chargers at every node for various range anxiety levels.

| Node | $m = 0$ | $m = 1$ | $m = 2$ | $m = 3$ | $m = 4$ |
|------|------|------|------|------|------|
| 1 | 0 | 0 | 0 | 0 | 0 |
| 2 | 0 | 0 | 0 | 0 | 0 |
| 3 | 0 | 0 | 0 | 0 | 0 |
| 4 | 0 | 0 | 0 | 0 | 0 |
| 5 | 0 | 0 | 4 | 0 | 0 |
| 6 | 0 | 0 | 0 | 0 | 4 |
| 7 | 0 | 4 | 0 | 4 | 0 |
| 8 | 0 | 0 | 0 | 2 | 0 |
| 9 | 0 | 2 | 2 | 2 | 2 |
| 10 | 0 | 0 | 0 | 0 | 0 |
| 11 | 5 | 0 | 0 | 0 | 0 |
| 12 | 0 | 2 | 2 | 0 | 2 |
| 13 | 0 | 0 | 0 | 0 | 0 |

Note: "$m$" denotes the range anxiety.

As seen in Figure 4, with the increase in the agents' range anxieties, the energy recharged increases where we find only agents of the OD pair $4 - 2$ need to recharge, and other agents do not need to recharge when the agents' range anxiety is set to 0, which is a reasonable phenomenon. As the agents' range anxiety increases, more agents need to recharge their BEV batteries, and the energy recharged for each OD pair increases. Furthermore, the total charging time increases, and the total trip time of all the agents will increase, as seen in Figure 5. As shown in Table 10, only one charging station is built when the range anxiety is set to 0, which reflects the same relationship.

To demonstrate the advantages of the agent-based model, we change the range anxiety levels of some agents of one OD pair to avoid homogeneous agents, and the specific changes are shown in Table 11. Then, the results of the total trip time are presented in Figure 6. As seen in Figure 6, the total time increases with the increasing proportion of agents at high range anxiety levels. This is a reasonable result. With increasing range anxiety, the total amount of charge will increase, and the total trip time further increases.

**Table 11.** Agent numbers with various range anxiety levels for one OD pair.

| OD | Case I (20%) | | Case II (40%) | | Case III (50%) | | Case IV (60%) | | Case V (80%) | |
|------|------|------|------|------|------|------|------|------|------|------|
| | $m = 2$ | $m = 3$ | $m = 2$ | $m = 3$ | $m = 2$ | $m = 3$ | $m = 2$ | $m = 3$ | $m = 2$ | $m = 3$ |
| $1 - 2$ | 16 | 4 | 12 | 8 | 10 | 10 | 8 | 12 | 4 | 16 |
| $1 - 3$ | 24 | 6 | 18 | 12 | 15 | 15 | 12 | 18 | 6 | 24 |
| $4 - 2$ | 24 | 6 | 18 | 12 | 15 | 15 | 12 | 18 | 6 | 24 |
| $4 - 3$ | 16 | 4 | 12 | 8 | 10 | 10 | 8 | 12 | 4 | 16 |

Note: "$m$" denotes the range anxiety.

*4.3. Various BEV Initial Charge Levels*

Another factor that may affect BEV driver's path choice is the initial power level of the vehicle. Various initial battery levels for BEVs with $L_0 = 14$, 16, 18, 22, and 24 kWh, in addition to 20 kWh are tested where the range anxiety values of all agents are set to 2 kWh. The test results of the energy recharged are shown in Figure 7, the total trip time results are shown in Figure 8, and the numbers of chargers built are shown in Table 12.

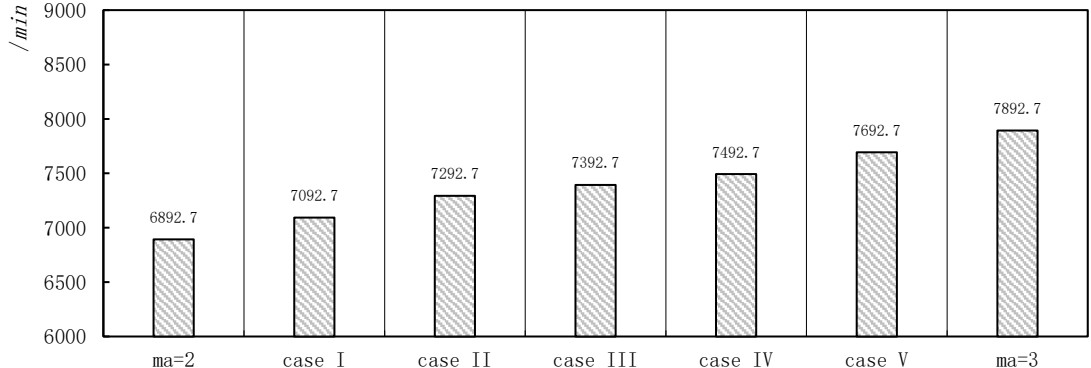

**Figure 6.** Comparison of the total trip time for various cases.

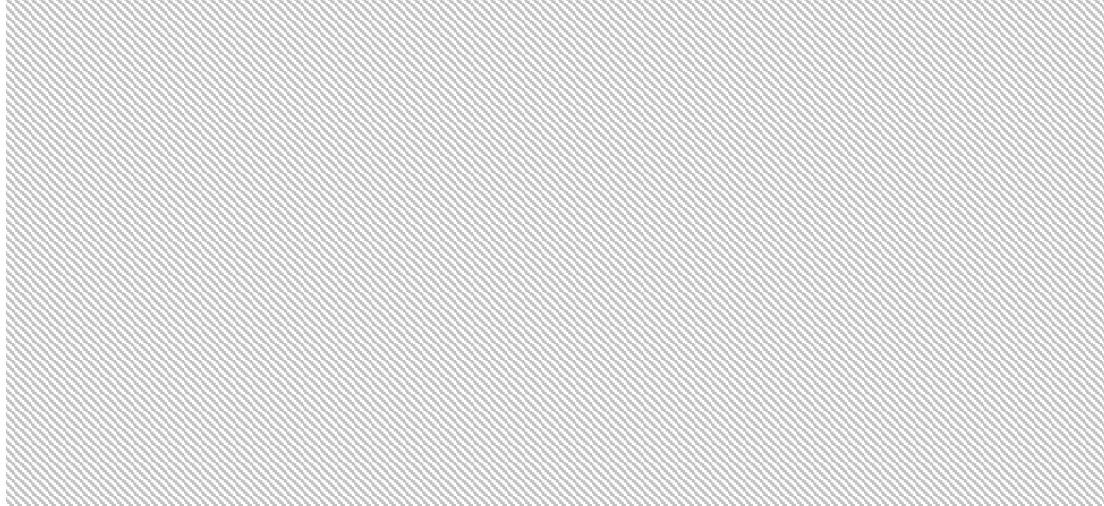

**Figure 7.** Comparison of the energy recharged for various BEV initial charge levels.

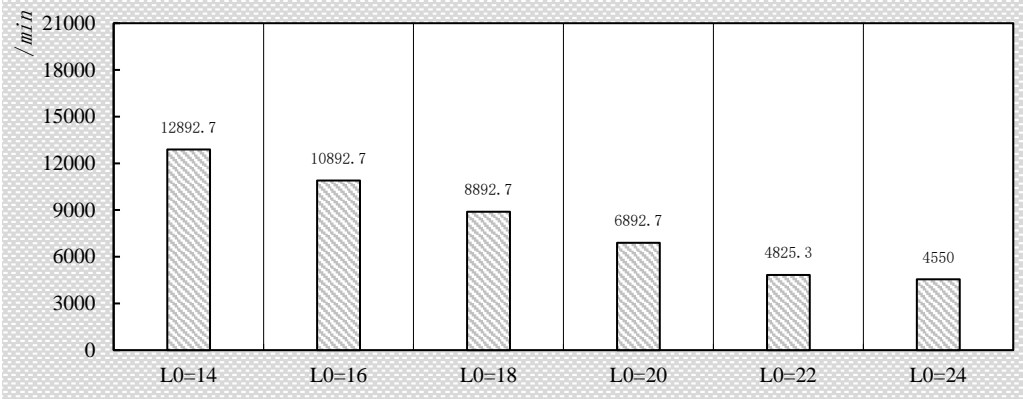

**Figure 8.** Comparison of the total trip time for various BEV initial charge levels.

As the initial charge level increases, both the amount recharged and the total time decrease, as seen in Figures 7 and 8, respectively. In Figure 7, the amount recharged is zero when $L_0 = 24$ kWh, suggesting that all BEV agents complete their trips with no need for recharging. Furthermore, we find that no charging station is built when $L_0 = 24$ kWh, and there is only one charging station built when $L_0 = 20$ kWh, as seen in Table 12, which is a reasonable charging station construction program when there is little or no demand for charging from all agents.

Similarly, we change the initial charge levels of some agents of one OD pair to avoid homogeneous agents, and examined results are shown in Table 13. Then, the results for the total trip time are presented in Figure 9. The total time decreases with an increasing proportion of agents with high initial charge

levels, presented in Figure 9, which is a reasonable result. With increasing initial charge levels, the total amount recharged will decrease, and the total trip time will decrease.

**Table 12.** Number of chargers at every node for various initial battery charge levels.

| Node | $L0 = 14$ | $L0 = 16$ | $L0 = 18$ | $L0 = 20$ | $L0 = 22$ | $L0 = 24$ |
|---|---|---|---|---|---|---|
| 1 | 0 | 0 | 0 | 0 | 0 | 0 |
| 2 | 0 | 0 | 0 | 0 | 0 | 0 |
| 3 | 0 | 0 | 0 | 0 | 0 | 0 |
| 4 | 0 | 0 | 0 | 0 | 0 | 0 |
| 5 | 4 | 0 | 0 | 4 | 0 | 0 |
| 6 | 0 | 4 | 4 | 0 | 0 | 0 |
| 7 | 0 | 0 | 0 | 0 | 0 | 0 |
| 8 | 0 | 0 | 0 | 0 | 0 | 0 |
| 9 | 2 | 2 | 2 | 2 | 0 | 0 |
| 10 | 0 | 0 | 0 | 0 | 0 | 0 |
| 11 | 0 | 0 | 0 | 0 | 5 | 0 |
| 12 | 2 | 2 | 2 | 2 | 0 | 0 |
| 13 | 0 | 0 | 0 | 0 | 0 | 0 |

**Table 13.** Numbers of agents with various initial charge levels for one OD pair.

| OD | Case I (20%) | | Case II (40%) | | Case III (50%) | | Case IV (60%) | | Case V (80%) | |
|---|---|---|---|---|---|---|---|---|---|---|
| | $L0 = 20$ | $L0 = 22$ | $L0 = 20$ | $L0 = 22$ | $L0 = 20$ | $L0 = 22$ | $L0 = 20$ | $L0 = 22$ | $L0 = 20$ | $L0 = 22$ |
| $1 - 2$ | 16 | 4 | 12 | 8 | 10 | 10 | 8 | 12 | 4 | 16 |
| $1 - 3$ | 24 | 6 | 18 | 12 | 15 | 15 | 12 | 18 | 6 | 24 |
| $4 - 2$ | 24 | 6 | 18 | 12 | 15 | 15 | 12 | 18 | 6 | 24 |
| $4 - 3$ | 16 | 4 | 12 | 8 | 10 | 10 | 8 | 12 | 4 | 16 |

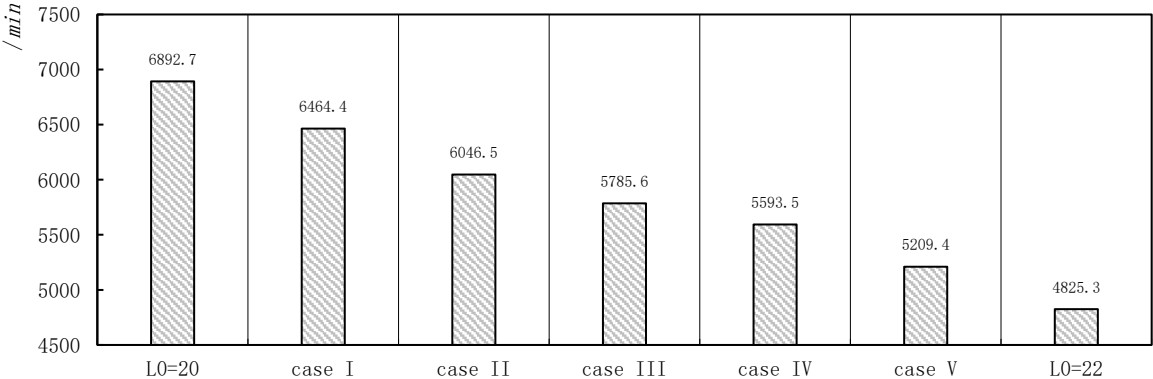

**Figure 9.** Comparison of the total trip time for various cases.

*4.4. Considering Different Total Cost Budgets*

The total cost budget can also affect the final solution, so the various total cost budgets considered are 48, 43, 33, and 27, in addition to 38, where the initial charge level equals to 20 kWh, and the range anxieties of all agents equal to 2 kWh. Figure 10 compares the energy recharged for all OD pairs. The amounts of energy recharged are almost equal when the total budget is 48, 43, or 38. Furthermore, the amounts recharged are almost equal when the total budget is 33 or 27. We can further observe that when the total budget $B$ is 33 or 27, the total trip times are greater than when $B$ is 38, 43, or 48. Moreover, agents of the OD pair $1 - 2$ do not need to recharge, while there is a substantial increase in the total recharge amount for the agents of OD pair $1 - 3$ when the total budget $B$ is 33 or 27. This is a reasonable result. With the reduced budget, the number of charging stations built is also reduced; thus, every link is at its flow capacity, so the number of agents that can be recharged is reduced.

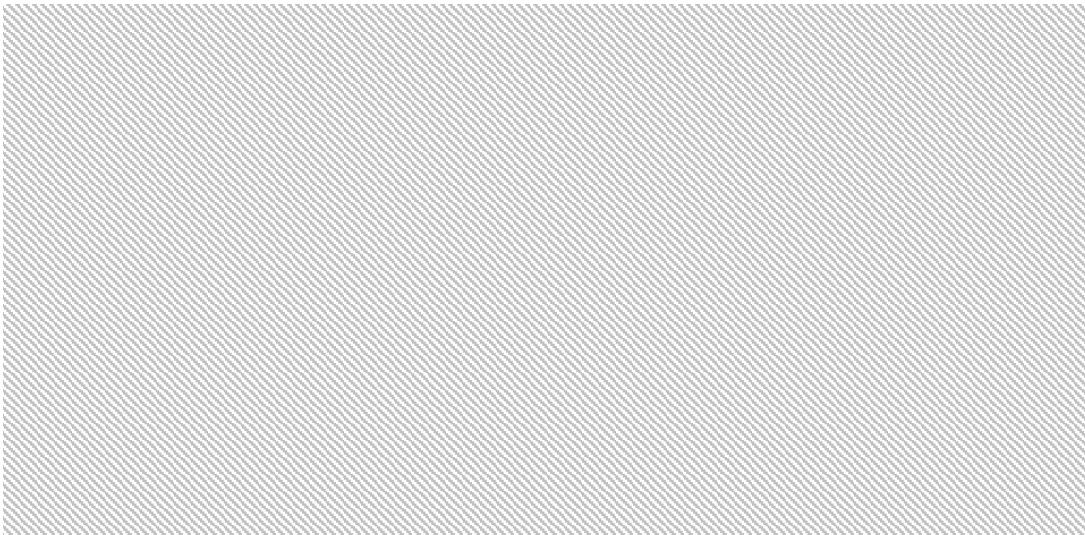

**Figure 10.** Comparison of the energy recharged for various total cost budgets.

Although the amounts of energy recharged are similar when the total budget is 48, 43, or 38, when the budget *B* is 48, three charging stations are constructed, and the number of chargers in each charging station is five; when the budget *B* is 38, three charging stations can be built as well, but there is only one charging station with four chargers and two charging stations with two chargers each, as shown in Table 14, which results in a longer queue time and a total increase in time, as seen in Figure 11.

**Table 14.** Number of chargers in each node for various total cost budgets.

| Node | *B* = 48 | *B* = 43 | *B* = 38 | *B* = 33 | *B* = 27 |
|:---:|:---:|:---:|:---:|:---:|:---:|
| 1 | 0 | 0 | 0 | 0 | 0 |
| 2 | 0 | 0 | 0 | 0 | 0 |
| 3 | 0 | 0 | 0 | 0 | 0 |
| 4 | 0 | 0 | 0 | 0 | 0 |
| 5 | 5 | 0 | 4 | 0 | 0 |
| 6 | 0 | 0 | 0 | 5 | 2 |
| 7 | 0 | 0 | 0 | 0 | 0 |
| 8 | 5 | 5 | 0 | 0 | 0 |
| 9 | 5 | 0 | 2 | 5 | 5 |
| 10 | 0 | 0 | 0 | 0 | 0 |
| 11 | 0 | 5 | 0 | 0 | 0 |
| 12 | 0 | 0 | 2 | 0 | 0 |
| 13 | 0 | 3 | 0 | 0 | 0 |

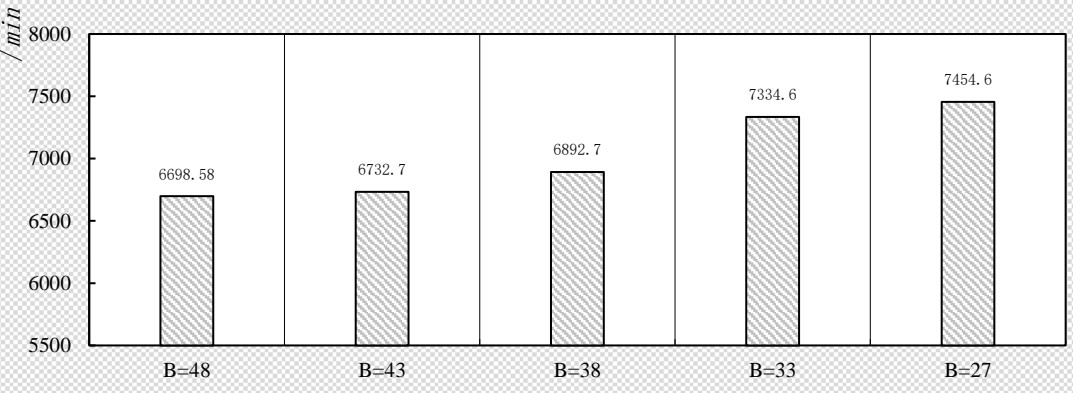

**Figure 11.** Comparison of the total trip time for various total cost budgets.

*4.5. A Larger Example*

In this section, a larger numerical example, named the Sioux Falls network (see Figure 12), is presented to examine the computational efficiency and its potential for application to real networks, we present numerical example. This network includes 24 nodes and 76 links. The characteristics are reported in Table 15, where the link distances are assumed to be 1.5 times the link travel time [8]. In this section, we choose eight OD pairs, and the OD demands are presented in Table 16. Parameter settings, such as the battery capacity $L_{max}$, the energy consumption rate $\overline{w}$, the range anxiety value $m^a$, and so forth, are the same as those mentioned in Section 4.1. The *Level* 2 charging level is also adopted. However, in this section, we assume the initial charge $L_0 = 0.2L_{max}$. We adopt the GAMS to solve the proposed model, and the equilibrium link flow is shown in Table 17. The construction program for charging stations is shown in Table 18.

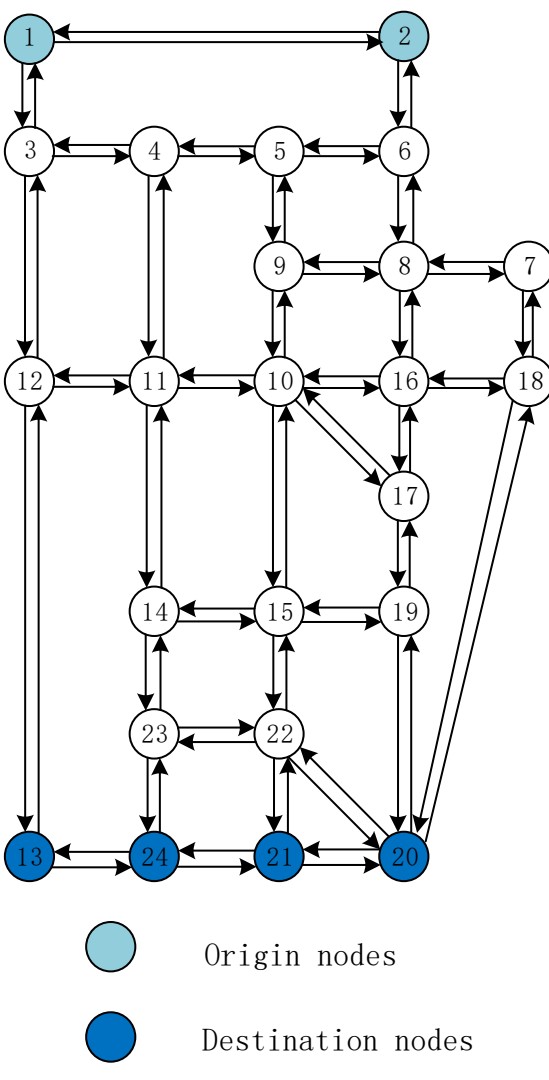

**Figure 12.** Sioux Falls network.

**Table 15.** Sioux Falls network information (link capacity (veh/h), distance (mi), travel time (min)).

| Link | Capacity | Distance | Travel Time | Link | Capacity | Distance | Travel Time |
|------|----------|----------|-------------|------|----------|----------|-------------|
| (1, 2) | 30 | 5.4 | 3.6 | (13, 24) | 45 | 3.6 | 2.4 |
| (1, 3) | 60 | 3.6 | 2.4 | (14, 11) | 20 | 3.6 | 2.4 |
| (2, 1) | 20 | 5.4 | 3.6 | (14, 15) | 20 | 4.5 | 3 |
| (2, 6) | 40 | 4.5 | 3 | (14, 23) | 25 | 3.6 | 2.4 |
| (3, 1) | 20 | 3.6 | 2.4 | (15, 10) | 25 | 5.4 | 3.6 |
| (3, 4) | 20 | 3.6 | 2.4 | (15, 14) | 20 | 4.5 | 3 |
| (3, 12) | 55 | 3.6 | 2.4 | (15, 19) | 20 | 3.6 | 2.4 |
| (4, 3) | 25 | 3.6 | 2.4 | (15, 22) | 20 | 3.6 | 2.4 |
| (4, 5) | 35 | 1.8 | 1.2 | (16, 8) | 20 | 4.5 | 3 |
| (4, 11) | 25 | 5.4 | 3.6 | (16, 10) | 30 | 4.5 | 3 |
| (5, 4) | 25 | 1.8 | 1.2 | (16, 17) | 20 | 1.8 | 1.2 |
| (5, 6) | 20 | 3.6 | 2.4 | (16, 18) | 20 | 2.7 | 1.8 |
| (5, 9) | 20 | 4.5 | 3 | (17, 10) | 25 | 6.3 | 4.2 |
| (6, 2) | 25 | 4.5 | 3 | (17, 16) | 25 | 1.8 | 1.2 |
| (6, 5) | 25 | 3.6 | 2.4 | (17, 19) | 30 | 1.8 | 1.2 |
| (6, 8) | 40 | 1.8 | 1.2 | (18, 7) | 20 | 1.8 | 1.2 |
| (7, 8) | 30 | 2.7 | 1.8 | (18, 16) | 30 | 2.7 | 1.8 |
| (7, 18) | 25 | 1.8 | 1.2 | (18, 20) | 20 | 3.6 | 2.4 |
| (8, 6) | 25 | 1.8 | 1.2 | (19, 15) | 20 | 3.6 | 2.4 |
| (8, 7) | 30 | 2.7 | 1.8 | (19, 17) | 20 | 1.8 | 1.2 |
| (8, 9) | 25 | 3 | 2 | (19, 20) | 30 | 3.6 | 2.4 |
| (8, 16) | 20 | 4.5 | 3 | (20, 18) | 20 | 3.6 | 2.4 |
| (9, 5) | 20 | 4.5 | 3 | (20, 19) | 20 | 3.6 | 2.4 |
| (9, 8) | 30 | 3 | 2 | (20, 21) | 25 | 5.4 | 3.6 |
| (9, 10) | 25 | 2.7 | 1.8 | (20, 22) | 20 | 4.5 | 3 |
| (10, 9) | 30 | 2.7 | 1.8 | (21, 20) | 25 | 5.4 | 3.6 |
| (10, 11) | 20 | 4.5 | 3 | (21, 22) | 20 | 1.8 | 1.2 |
| (10, 15) | 20 | 5.4 | 3.6 | (21, 24) | 30 | 2.7 | 1.8 |
| (10, 16) | 25 | 4.5 | 3 | (22, 15) | 20 | 3.6 | 2.4 |
| (10, 17) | 20 | 6.3 | 4.2 | (22, 20) | 20 | 4.5 | 3 |
| (11, 4) | 20 | 5.4 | 3.6 | (22, 21) | 25 | 1.8 | 1.2 |
| (11, 10) | 20 | 4.5 | 3 | (22, 23) | 20 | 3.6 | 2.4 |
| (11, 12) | 30 | 5.4 | 3.6 | (23, 14) | 30 | 3.6 | 2.4 |
| (11, 14) | 30 | 3.6 | 2.4 | (23, 22) | 20 | 3.6 | 2.4 |
| (12, 3) | 30 | 3.6 | 2.4 | (23, 24) | 30 | 1.8 | 1.2 |
| (12, 11) | 20 | 5.4 | 3.6 | (24, 13) | 20 | 3.6 | 2.4 |
| (12, 13) | 60 | 2.7 | 1.8 | (24, 21) | 30 | 2.7 | 1.8 |
| (13, 12) | 35 | 2.7 | 1.8 | (24, 23) | 25 | 1.8 | 1.2 |

**Table 16.** OD demand.

| O/D | 13 | 24 | 21 | 20 |
|-----|----|----|----|----|
| 1 | 10 | 12 | 10 | 15 |
| 2 | 10 | 15 | 10 | 10 |

The percentages of the frequencies of each charging station are selected and amounts of energy recharged at each charging station are reported in Figures 13 and 14, respectively. As seen in Table 18, node 12 is the largest charging station, which contains four chargers. Therefore, agents are more willing to recharge their batteries at this charging station due to the minimal queue time, which can be observed in Figure 13. This is a reasonable result. However, it is also reasonable to observe from Figure 14 that the amount of energy recharged at node 6 is greater than that at node 12. Because of the problems discussed in this model, the charging rates for the charging stations are the same, and the differences between the charging stations of different sizes lie in the queue time. Furthermore, owing to the initial charge level, where a full battery is not assumed for agents starting from their origin

nodes, we find that there is only one built charging station at node 1, the origin node reasonably; in reality, not all of the starting places are the drivers' own homes or companies with charging facilities.

**Table 17.** Equilibrium link flow.

| Link | Link Flow | Link | Link Flow | Link | Link Flow |
|------|-----------|------|-----------|------|-----------|
| (1,2) | 0 | (10,11) | 1 | (17,16) | 0 |
| (1,3) | 53 | (10,15) | 2 | (17,19) | 10 |
| (2,1) | 5 | (10,16) | 0 | (18,7) | 0 |
| (2,6) | 40 | (10,17) | 0 | (18,16) | 10 |
| (3,1) | 1 | (11,4) | 1 | (18,20) | 19 |
| (3,4) | 0 | (11,10) | 0 | (19,15) | 0 |
| (3,12) | 53 | (11,12) | 4 | (19,17) | 0 |
| (4,3) | 0 | (11,14) | 0 | (19,20) | 11 |
| (4,5) | 0 | (12,3) | 1 | (20,18) | 0 |
| (4,11) | 4 | (12,11) | 0 | (20,19) | 0 |
| (5,4) | 4 | (12,13) | 60 | (20,21) | 19 |
| (5,6) | 0 | (13,12) | 4 | (20,22) | 2 |
| (5,9) | 0 | (13,24) | 42 | (21,20) | 15 |
| (6,2) | 4 | (14,11) | 0 | (21,22) | 0 |
| (6,5) | 4 | (14,15) | 0 | (21,24) | 14 |
| (6,8) | 32 | (14,23) | 2 | (22,15) | 1 |
| (7,8) | 29 | (15,10) | 0 | (22,20) | 1 |
| (7,18) | 0 | (15,14) | 1 | (22,21) | 3 |
| (8,6) | 0 | (15,19) | 1 | (22,23) | 0 |
| (8,7) | 29 | (15,22) | 1 | (23,14) | 0 |
| (8,9) | 3 | (16,8) | 0 | (23,22) | 2 |
| (8,16) | 0 | (16,10) | 0 | (23,24) | 2 |
| (9,5) | 0 | (16,17) | 10 | (24,13) | 2 |
| (9,8) | 0 | (16,18) | 0 | (24,21) | 27 |
| (9,10) | 3 | (17,10) | 0 | (24,23) | 2 |
| (10,9) | 0 | | | | |

**Table 18.** Charging station construction program.

| Node | Charger Number (char) |
|------|------------------------|
| 1 | 2 |
| 6 | 2 |
| 12 | 4 |

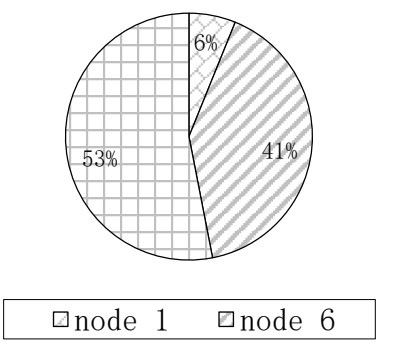

**Figure 13.** Percentages of the frequencies of each charging station selected.

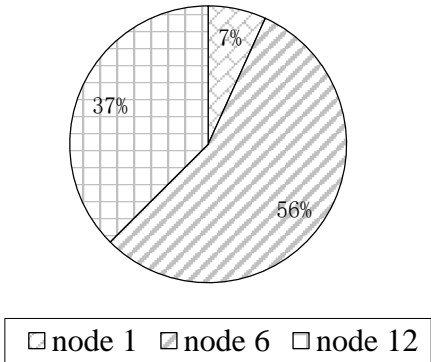

**Figure 14.** Percentages of the amounts of energy recharged at each charging station.

## 5. Conclusions and Discussions

There is a strong demand for the deployment of charging stations, including from policy makers. The policy maker not only needs to determine the location of charging stations, but also the number of chargers that each charging station should be given to create an effective allocation of resources. Therefore, this paper has conducted in-depth research on the location of charging stations, and also discussed the number of chargers. Specifically, firstly, our research presents a novel approach to deployment locations for BEV charging stations. Previously, charging stations of the same size are discussed. Furthermore, multiple types of charging stations with different charging speeds have been studied [22], while the size of charging stations has not been the focus of previous studies. There are charging stations of different sizes with different charging service efficiencies in reality. Therefore, our research on various charging station sizes addresses an important gap in previous research. Secondly, in our study, an agent-based location problem for charging stations for BEVs is adopted instead of a traditional flow-based model. Therefore, the agent differences of one OD pair can be reflected. Thirdly, an MILP formulation of the ARMSLP-CN is presented. Finally, using the GAMS commercial solver, the problem could be solved directly. To demonstrate the model, the Nguyen-Dupius and Sioux Falls networks are solved and sensitivity analyses regarding range anxiety, initial state of charge, total cost budget, and different charging speeds depending on the type of charging devices, are also conducted.

Furthermore, there are some potential extended ways forward for this model. First, an efficient algorithm is expected, because variables will increase rapidly in the real network, which make the efficiency of solvers seriously insufficient. Moreover, more realistic conditions, such as weather, temperature, and so on, are worth considering. We will further increase the application of real data. Paul Brooker and Qin [34] used real data to analyze the potential location of charging stations to increase the service efficiency of charging devices. Therefore, we should further apply the model to the real network. Furthermore, Oda et al., [35,36] used the historical data of current charging equipment used to optimize the number of chargers that should be further increased in the existing charging stations with the aim of minimum charging waiting time. We should optimize the number of chargers on the basis of the existing chargers, which can avoid the waste of original resources.

**Author Contributions:** Formal analysis, S.C.; Methodology, S.C.; Writing—original draft, S.C.; Funding acquisition, H.Z.; Supervision, H.Z.; Writing—review and editing, C.Z.

**Acknowledgments:** This work is jointly supported by the National Natural Science Foundation of China (71371028, 71621001) and the Research Foundation of BISTU (1825028).

**Conflicts of Interest:** The authors declare no conflict of interest.

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
