# Peer review of "Locating Charging Stations of Various Sizes with Different Numbers of Chargers for Battery Electric Vehicles"

_energies, doi:10.3390/en11113056_

Round 1
Reviewer 1 Report
This topic is interesting and relevant to Energies journal on locating charging stations of various sizes in a 13 nodes Nguyen Dupius network and 24X76 Sioux Falls network. The article has proposed methodology using agent-based integer linear formulation which is innovative in combining behavioural simulation with deterministic optimization methodology i.e. mixed integer programming.
The following are peer review comments to the authors.
· In p3, queue time is introduced as one of the three components of BEV charging time and these components were further elaborated in Section 2.2. It could be easier for the reader if this phrase is defined (briefly) in the first use or specify in brackets Section 2.2 for more details.
· In p8, it is unclear what does “char” mean (e.g. Z2 is 4 char). It has been repeatedly used in that paragraph.
· In p11, what units are for total budget (38), fixed construction cost (10) and construction cost of one charger (1)? Are these costs in proportion to what was observed in real-life?
· With the proposed Agent-based integer linear formulation using Ngugen Dupius network as a case study problem, the following sensitivities have been studied and reported a) Range anxiety b) initial charge levels.
· Sec 2.2 assumed fixed/constant linear recharge rate (III). What is the implication of this assumption on the proposed model? Discuss limitation.
· Whether choice of Nguyen Dupius and Sioux Falls networks has implication on generalizability of findings from the proposed approach? Discuss limitation.
· Overall, the manuscript is written in a simple and easy to read style.
In general, the proposed method is interesting and relevant to policy makers in an emerging industry. The article could be accepted for publications with minor amends.
Reviewer 2 Report
The paper is well written and offers interesting findings. Several major issues must be improved by the authors:
Many references are lumped without giving sufficient description for each of them. Avoid this kind of writing and provide a brief description for each of them. Therefore, the readers can understand easily what the references are pointing at.
Additional references focusing on the charger's distribution must be performed to increase and clarify the novelty of the study. Below are several literatures which have been found.
T Oda, M Aziz, T Mitani, Y Watanabe, T Kashiwagi. Mitigation of congestion related to quick charging of electric vehicles based on waiting time and cost–benefit analyses: A japanese case study. Sustainable cities and society 36, 99-106 (2016).
Jia He, Hai Yang, Tie-Qiao Tang, Hai-Jun Huang. An optimal charging station location model with the consideration of electric vehicle’s driving range. Transportation Research Part C: Emerging Technologies, 86, 641-654 (2018).
Xiaomin Xi, Ramteen Sioshansi, Vincenzo Marano. Simulation–optimization model for location of a public electric vehicle charging infrastructure. Transportation Research Part D: Transport and Environment, 22, 60-69 (2013).
T Oda, M Aziz, T Mitani, Y Watanabe, T Kashiwagi. Actual Congestion and Effect of Charger Addition in the Quick Charger Station: Case Study Based on the Records of Expressway. Electrical Engineering in Japan 198 (2), 11-18 (2017)
R. Paul Brooker, Nan Qin. Identification of potential locations of electric vehicle supply equipment. Journal of Power Sources, 299, 76-84 (2015)
Charging is not always conducted with a certain rated capacity, however, it really depends on several conditions, including SoC, temperature, etc. Do you consider these factors in your study?
Author Response
Additional references focusing on the charger's distribution must be performed to increase and clarify the novelty of the study. Below are several literatures which have been found.
R. Paul Brooker, Nan Qin. Identification of potential locations of electric vehicle supply equipment. Journal of Power Sources, 299, 76-84 (2015)
T Oda, M Aziz, T Mitani, Y Watanabe, T Kashiwagi. Mitigation of congestion related to quick charging of electric vehicles based on waiting time and cost–benefit analyses: A japanese case study. Sustainable cities and society 36, 99-106 (2016).
T Oda, M Aziz, T Mitani, Y Watanabe, T Kashiwagi. Actual Congestion and Effect of Charger Addition in the Quick Charger Station: Case Study Based on the Records of Expressway. Electrical Engineering in Japan 198 (2), 11-18 (2017)
The above three papers are based on data. The first paper uses data to determine the potential location of charging stations. The latter two papers use the true data to determine the number of chargers that should be added on the original basis. We think that these papers are good research direction, so they are quoted and discussed in detail in the conclusion section.
Jia He, Hai Yang, Tie-Qiao Tang, Hai-Jun Huang. An optimal charging station location model with the consideration of electric vehicle’s driving range. Transportation Research Part C: Emerging Technologies, 86, 641-654 (2018).
In this paper, the electric vehicle’s driving range is considered, while the related paper, Lee and Han (2017) has been cited and discussed, so the related research above is not cited again.
Lee, C., Han, J., 2017. Benders-and-Price approach for electric vehicle charging station location problem under probabilistic travel range. Transp. Res. Part B 106,130-152.
Xiaomin Xi, Ramteen Sioshansi, Vincenzo Marano. Simulation–optimization model for location of a public electric vehicle charging infrastructure. Transportation Research Part D: Transport and Environment, 22, 60-69 (2013).
For this paper, we do not cite it, because the paper develops a simulation–optimization model that determines where to locate electric vehicle chargers to maximize their use by privately owned electric vehicles. We think this is a problem of multiple types of recharging stations where there are two level chargers considered, while the related research of Wang and Lin (2013) has discussed more types and has been cited. Therefore, we do not cite this paper.
Wang, Y.W., Lin, C.C., 2013. Locating multiple types of recharging stations for battery-powered electric vehicle transport. Transp. Res. Part E 58, 76-87.
Charging is not always conducted with a certain rated capacity, however, it really depends on several conditions, including SoC, temperature, etc. Do you consider these factors in your study?
There are many factors that affect charging and power consumption, such as: weather, temperature, speed, which will be a complicated study. In Section 2.2 of this article, we make assumptions about related problems. For example, charging rate is constant, and only depends on the type of charging devices, so in the data analysis of this paper, the three devices with different charging rates are discussed. And then we assume that the rate of power consumption is constant, and the total power consumption is only related to the total running distance. Gradient and overcrowded links significantly affect a BEV’s speed. Some conditions, such as weather, temperature, speed, may change the travel ranges of BEVs up to 60% [1]. Straubel [2] reported that the travel range of Tesla, a maker of BEVs, may be reduced by half when the driver doubles the speed. Therefore, the uncertain travel range is a complex study where Lee and Han [3] established the problem in a probabilistic consideration. In the future, we will increase research related to realistic conditions.
[1] Neubauer, J., Wood, E., Pesaran, A., 2013. Project Milestone. Analysis of Range Extension Techniques for Battery Electric Vehicles. Technical Report. National Renewable Energy Laboratory (NREL), Golden, CO (United States).
[2] Straubel, J., 2015. Driving range for the model S family.
[3] Lee, C., Han, J., 2017. Benders-and-Price approach for electric vehicle charging station location problem under probabilistic travel range. Transp. Res. Part B 106,130-152.

Reviewer 3 Report
1-Please explain with more details what is the difference between your woark and other work which you have published in:
Locating Multiple Size and Multiple Type of Charging Station for Battery Electricity Vehicles
by Shaohua Cui,Hui Zhao,Huijie Wen andCuiping Zhang
Sustainability 2018, 10(9), 3267; https://doi.org/10.3390/su10093267
2- Why we need this kind of complexity in planning of charging station?
3- The novelty of work is not clear.
Author Response
1-Please explain with more details what is the difference between your work and other work which you have published in:
Locating Multiple Size and Multiple Type of Charging Station for Battery Electricity Vehicles
by Shaohua Cui,Hui Zhao,Huijie Wen andCuiping Zhang
Sustainability 2018, 10(9), 3267; https://doi.org/10.3390/su10093267
We are the first to study the size of charging stations and determine the number of chargers. Because if the designer can deploy the number of chargers when they are planning the location of charging stations, the total resources are properly allocated, therefore, we think this will make resources fully be utilized and we should advance research in related directions.
The main contribution of the published paper mentioned by the reviewer is the deployment of multiple types of charging devices with different charging speeds. Although the number of chargers is mentioned and discussed in the article, due to the discussion of multiple types of charging devices, the model focuses on the conversion of cross-multiplication part, which makes the discussion of the number of chargers be ignored. In other words, the published paper is only an extension of the paper.
In this paper, we use a lot of space to write the problem of the size of charging stations, including the reason for different charging time caused by charging stations of different sizes, and the calculation of queuing time, which is the core of this paper.
A large charging station with more chargers can simultaneously meet more charging requests of battery electric vehicles, and the queuing time is less. On the contrary, charging stations with fewer chargers have poor service efficiency, and therefore the drivers of battery electric vehicles will spend more queuing time. For the processing of queuing time, we even add some intermediate variables, which are described in detail in Section 2.3.
And in this paper, in order to reflect the difference in the queuing time of charging stations with the different number of chargers, we design a small numerical network with three paths in Section 2.4 to explain and analyze the related calculation. Because this article is the first study of the size of charging stations, we do not have the relevant cost data, such as total budge, fixed construction cost and construction cost of one charger. And then we can only assume the value of the three parameters according to previous studies. Similarly, in the related extended papers that have been published, we still take the assumed data. Although in the part of results we conducted sensitivity analyses, however, the impact of the size of charging stations has been ignored to some extent, so in this article, we remove the irrelevant variables, and only the size of the charging station is studied in depth. In the section of results, sufficient sensitivity analyses are conducted to provide a large amount of data for other related later research.
2- Why we need this kind of complexity in planning of charging station?
There is a strong demand for the deployment of charging stations, such as policy makers. And the policy maker not only needs to determine the location of the charging station, but also the number of chargers for each charging station should be given to make the effective allocation of resources. Therefore, this paper has conducted in-depth research on the location of charging stations, and also discusses the number of chargers. Chen et al., (2016) studied the problem of the deployment of charging lanes, and in their study, and the length and the location of the charging lane are given accurately. However, for the plug-in charger, there is no research on the number of chargers. There are charging stations of different sizes with different charging service efficiencies in reality absolutely. Therefore, our research on various charging station sizes addresses an important gap in previous research.
Chen, Z.B., He, F., Yin, Y.F., 2016. Optimal deployment of charging lanes for electric vehicles in transportation networks. Transp. Res. Part B 91, 344-365.
Reviewer 4 Report
The subject of manuscript is interesting. The technical content of the paper is acceptable and contribution is clear. The idea of locating charging of charging stations with vaiable sizes and ratings is interesting however, the following comments are suggested to the authors:
- the title is a bit strange and needs modifications. It is not professional to have repetitive words in especially in the title. The authors are asked to remove one of the words “ various “ in the title.
- Again, in some sections, some statements have been repeated and need to be modified. For example, in section 2 “ problem formulation “, this statement is repeated “ we are the first to study”. This section needs to start in another way with other wording as this statement has already been used in defining the second contribution.
- In page 11, how the fixed construction cost of one charging station is defined in this study? How about the construction cost for one charger? Are these practical and based on any resources?
- The conclusion is also suggested to be modified to be in the form of dot points explaining the contributions of paper.
Author Response
1) the title is a bit strange and needs modifications. It is not professional to have repetitive words in especially in the title. The authors are asked to remove one of the words “ various “ in the title.
Thanks to the reviewer's suggestion, we decided to modify the repeated words. The revised title is as follows:
Locating charging stations of various sizes with different numbers of chargers for battery electric vehicles
2) Again, in some sections, some statements have been repeated and need to be modified. For example, in section 2 “ problem formulation “, this statement is repeated “ we are the first to study”. This section needs to start in another way with other wording as this statement has already been used in defining the second contribution.
Thanks to the reviewer's comments, we have modified the text to avoid excessive repeated statements.
3) In page 11, how the fixed construction cost of one charging station is defined in this study? How about the construction cost for one charger? Are these practical and based on any resources?
These three parameters have no clear units because they are only proportional. Because there are no accurate values of these parameters for our research, we have adjusted and assumed the proportions according to the previous study of Liu and Wang (2017).
Liu, H., Wang, D.Z.W., 2017. Locating multiple types of charging facilities for battery electric vehicles. Transp. Res. Part B 103, 30-55.
4) The conclusion is also suggested to be modified to be in the form of dot points explaining the contributions of paper.
Thanks to the reviewer's comments, we believe that the writing form of dot points can be easily read by the readers, so we have revised the conclusions. The revised text is as follows:
There is a strong demand for the deployment of charging stations, such as policy makers. And the policy maker not only needs to determine the location of the charging station, but also the number of chargers for each charging station should be given to make the effective allocation of resources. Therefore, this paper has conducted in-depth research on the location of charging stations, and also discusses the number of chargers. Specifically, firstly, our research presents a novel approach to deploy locations for BEV charging stations. Previously, charging stations of the same size are discussed. Furthermore, multiple types of charging stations with different charging speeds have been studied [22], while the size of charging stations has not been the focus of previous studies. There are charging stations of different sizes with different charging service efficiencies in reality absolutely. Therefore, our research on various charging station sizes addresses an important gap in previous research. Secondly, in our study, an agent-based location problem for charging stations for BEVs is adopted instead of a traditional flow-based model. Therefore, the agent differences of one OD pair can be reflected. Thirdly, an MILP formulation of the ARMSLP-CN is presented. Finally, using the GAMS commercial solver, the problem could be solved directly. To demonstrate the model, the Nguyen–Dupius and Sioux Falls networks are solved and sensitivity analyses regarding range anxiety, initial state of charge, total cost budget, and different charging speeds depending on the type of charging devices, are also conducted.

Round 2
Reviewer 1 Report
Revised manuscript looks fine to be published
Reviewer 2 Report
The authors have sufficiently improved and corrected the manuscript. I think it is acceptable for publication in the current form.
Reviewer 3 Report
All comments have been done by authors.
Reviewer 4 Report
Authors addressed all the comments.